# Quantifying Parkinson's disease severity using mobile wearable devices and machine learning: the ParkApp pilot study protocol

Gent Ymeri [1,2] Dario Salvi,[1,2] Carl Magnus Olsson,[1,2]
Myrthe Vivianne Wassenburg,[3,4] Athanasios Tsanas,[5,6] Per Svenningsson[3,4]

¹Department of Computer Science and Media Technology (DVMT), Malmö University, Malmö, Sweden
²Internet of Things and People Research Center (IOTAP), Malmö University, Malmö, Sweden
³Department of Clinical Neuroscience, Karolinska Institute, Stockholm, Sweden
⁴Center for Neurology, Academic Specialist Center Torsplan, Region Stockholm, Sweden
⁵Usher Institute, Edinburgh Medical School, The University of Edinburgh, Edinburgh, UK
⁶Alan Turing Institute, London, UK

**Correspondence to**
Gent Ymeri; gent.ymeri@mau.se

## ABSTRACT

**Introduction** The clinical assessment of Parkinson's disease (PD) symptoms can present reliability issues and, with visits typically spaced apart 6 months, can hardly capture their frequent variability. Smartphones and smartwatches along with signal processing and machine learning can facilitate frequent, remote, reliable and objective assessments of PD from patients' homes.
**Aim** To investigate the feasibility, compliance and user experience of passively and actively measuring symptoms from home environments using data from sensors embedded in smartphones and a wrist-wearable device.
**Methods and analysis** In an ongoing clinical feasibility study, participants with a confirmed PD diagnosis are being recruited. Participants perform activity tests, including Timed Up and Go (TUG), tremor, finger tapping, drawing and vocalisation, once a week for 2 months using the Mobistudy smartphone app in their homes. Concurrently, participants wear the GENEActiv wrist device for 28 days to measure actigraphy continuously. In addition to using sensors, participants complete the Beck's Depression Inventory, Non-Motor Symptoms Questionnaire (NMSQuest) and Parkinson's Disease Questionnaire (PDQ-8) questionnaires at baseline, at 1 month and at the end of the study. Sleep disorders are assessed through the Parkinson's Disease Sleep Scale-2 questionnaire (weekly) and a custom sleep quality daily questionnaire. User experience questionnaires, Technology Acceptance Model and User Version of the Mobile Application Rating Scale, are delivered at 1 month. Clinical assessment (Movement Disorder Society-Unified Parkinson Disease Rating Scale (MDS-UPDRS)) is performed at enrollment and the 2-month follow-up visit. During visits, a TUG test is performed using the smartphone and the G-Walk motion sensor as reference device. Signal processing and machine learning techniques will be employed to analyse the data collected from Mobistudy app and the GENEActiv and correlate them with the MDS-UPDRS. Compliance and user aspects will be informing the long-term feasibility.
**Ethics and dissemination** The study received ethical approval by the Swedish Ethical Review Authority (Etikprövningsmyndigheten), with application number 2022-02885-01. Results will be reported in peer-reviewed journals and conferences. Results will be shared with the study participants.

## STRENGTHS AND LIMITATIONS OF THIS STUDY

⇒ This study uses a mobile and wearable device to conveniently, non-invasively, passively and actively quantify symptoms at home in a user-friendly manner.
⇒ The study combines objective data collected across multiple tests, to evaluate the frequency and severity of motor and non-motor symptoms.
⇒ The study aims to identify the link between data collected at home and conventional clinical scales, potentially enhancing the remote monitoring and management of Parkinson's disease.
⇒ This study excludes participants with severe symptoms who are unable to use smartphones, which may result in a less diverse sample and limit the generalisability of the findings.
⇒ Data analysis is dependent on the quantity and variability of collected data; success of the study relies on participants' compliance with the intensive data collection regime.

## INTRODUCTION

Parkinson's disease (PD) is a heterogeneous neurodegenerative disorder characterised by third cardinal symptoms: bradykinesia, resting tremor and musculoskeletal rigidity as well as several non-motor symptoms including cognitive alterations, anxiety, depression, hallucinations, pain, speech issues, sleep disorders along with autonomic disorders such as bladder and bowel problems and orthostatic hypotension.[1] PD is the second most common neurodegenerative disorder, with a worldwide prevalence estimated to reach 13 million cases by 2040.[2] The diagnosis and assessment of PD are mainly based on clinical findings such as history and physical examinations.[3] The most used and recommended rating scales to measure PD disease severity, progression and functional disability are the Hoehn and Yahr staging scale and the Movement Disorder Society-Unified Parkinson

Disease Rating Scale (MDS-UPDRS).[4 5] The MDS-UPDRS is usually performed twice a year during clinical appointments to track disease progression. Both scales can present reliability issues, use ordinal scores and do not track disease burden in a home environment at regular intervals.[6]

To complement in-clinic examinations, patients may be asked to record their symptoms in diaries at home; however, these diaries often lack the necessary accuracy[7] needed to effectively supplement clinical assessments and are prone to recall bias. Smartphones offer a more convenient alternative to pen and paper, allowing for more efficiency and accuracy while minimising such biases.[8] Nevertheless, self-reports alone are insufficient to capture the full extent of symptoms, and additional technologies have been demonstrated to provide significant benefits across a range of applications.[9] Consequently, there is a need for systems that allow PD patients to be conveniently, reliably, objectively and longitudinally monitored in their home environments.

Accelerometers, particularly those embedded in mobile phones and smartwatches, have recently been explored as cost-effective, user-friendly solutions to monitor PD symptoms, and facilitating better follow-up, care and improved patient management.[10–12] As an example, Lopez-Blanco and coworkers[13] explore the gyroscope embedded in smartwatches to analyse rest tremor in PD patients. Their results show that using smartwatches for assessing rest tremor can correlate with clinical score and is well accepted by users. Czech and coworkers[14] investigate wearable devices to measure gait in PD. Their results prove that measuring gait in PD can be performed reliably with a single accelerometer. However, their study uses a lumbar-mounted accelerometer and does not investigate common wearable devices such as smartwatches or smartphones. Some research projects extend the range of monitored symptoms, such as in[15] where the authors investigate the quantification of dexterity in PD through a smartphone-based system. In this paper, authors incorporate finger tapping (FT) and drawing tests in a clinical trial including 19 PD participants and 22 healthy controls. Their results show weak to moderate correlations between UPDRS items and smartphone-based ratings; however, this study was performed in a clinical environment and does not show that the same quality of data can be obtained at home. In addition, the more recent mPower study[16] explores assessment of PD using a smartphone application in home environments. Recruitment of participants in this study was performed online which did not guarantee that all the participants were clinically diagnosed with PD. This study incorporates a number of tasks to gather data with the participants' mobile phones, including FT, vocalisation exercises, walk, balance and memory tests. Their results show that the FT task was more predictive for the self-reported status of PD. A smartphone app, cloudUPDRS app, is also introduced in[17] which also includes a number of data collection activities such as reaction, tremor and walking tests, and a well-being test, but excluding voice or cognition tests.

Despite the existence of various digital health tools for PD monitoring, these typically focus on a single measurement modality, or the recruitment of the participants does not generalise the results for at-home assessments. A multimodal, user-friendly and affordable solution for home-based continuous monitoring of PD could provide significant advantages for both patients and clinicians.

Considering this, the primary aim of this study is to determine the feasibility of reliably measuring PD symptoms using mobile phones and wearable devices at home and to provide a more holistic overview of symptomatology compared with what has been attempted in previous studies. The study is supported by the development of a novel, multimodal symptoms tracker system, which both passively and actively quantify motor symptoms such as bradykinesia, gait impairment, tremor, voice disorders, as well as non-motor symptoms like depression, and sleep disturbances.

## METHODS AND ANALYSIS
### Study design and participants
This clinical feasibility study involves the recruitment of 30 participants with clinically diagnosed PD at the Center for Neurology, Academic Specialist Center Torsplan (ASCT), Stockholm Health Services, Region Stockholm.

Participants are recruited based on the inclusion and exclusion criteria summarised in table 1. Guided by similar studies that have used between 10 and 50 participants,[18–20] the goal of this study is towards hypothesis generation and assessing feasibility. Efforts are made to recruit participants with a diverse range of PD symptoms while avoiding over-representation of one sex or age category. Given the limited number of participants and the exploratory nature of this study, no specific software is used for randomisation; an ad hoc approach is adopted depending on the circumstances. Recruitment started in the fourth quarter of 2022, with plans to complete data collection in the fourth quarter of 2023.

**Table 1** Inclusion/exclusion criteria

| Inclusion criteria | Exclusion criteria |
|---|---|
| Must be diagnosed with Parkinson's disease and under treatment at Karolinska University Hospital or Academic Specialist Center Torsplan | Not able to use a smartphone independently |
| Must have mild or moderate motor and non-motor symptoms | Unable to read and understand Swedish |
| Must own a smartphone, Android or iPhone | Severe symptoms (H&Y 3 and above) |
| H&Y, Hoehn and Yahr. | |

**Table 2** The schedule for when activity tasks and questionnaires are performed by the recruited participants

| Activity tasks and questionnaires | Day after recruitment | | | | | | | |
|---|---|---|---|---|---|---|---|---|
| Timed Up and Go test | 1 | 8 | 15 | 22 | 29 | 36 | 43 | 50 |
| Beck's Depression Inventory | 3 | 31 | 54 | | | | | |
| Parkinson's Disease Sleep Scale | 7 | 14 | 21 | 28 | 35 | 42 | 49 | 56 |
| NMSQuest | 3 | 31 | 54 | | | | | |
| PDQ-8 | 3 | 31 | 54 | | | | | |
| Vocalisation test | 1 | 8 | 15 | 22 | 29 | 36 | 43 | 50 |
| Hold your phone test | 1 | 8 | 15 | 22 | 29 | 36 | 43 | 50 |
| Finger tapping test | 1 | 8 | 15 | 22 | 29 | 36 | 43 | 50 |
| Drawing test | 1 | 8 | 15 | 22 | 29 | 36 | 43 | 50 |
| Sleep Quality Questionnaire | Daily | | | | | | | |
| uMARS | 20 | | | | | | | |
| TAM | 20 | | | | | | | |
| Actigraphy | Continuously, recording at 25 Hz until battery is depleted after 28 days | | | | | | | |

NMSQuest, Non-Motor Symptoms Questionnaire; PDQ-8, Parkinson's Disease Questionnaire; TAM, Technology Acceptance Model; uMARS, User Version of the Mobile Application Rating Scale.

## Outcomes

The research questions that this study addresses are:

1. Are the multimodal measurements collected by mobile phones and wearable devices valid for estimating PD symptoms?
2. What extracted information/features from raw data are predictive of MDS-UPDRS scores using data collected by mobile phones and wearables?
3. Are mobile phones and wearable devices usable and accepted for data collection?

These research questions entail collecting data from devices at participants' home and reference data in-clinic and developing algorithms to associate the MDS-UPDRS score from the data collected at home. The questions also imply measuring compliance to the data regime protocol, usability of the technology and long-term acceptance.

## Patient and public involvement

None.

## Data collection

Data from participants are collected through two clinical visits—at baseline and after 2 months—as well as in between visits by using a mobile phone app and a wearable device in home environment. The mobile phone app used in this study is Mobistudy,[21] a free and open-source app designed for clinical research. The app prompts users to perform scheduled tasks during their 2-month participation in the study, as outlined in table 2. Tasks include answering questionnaires and performing motor tests. The motor tests are prompted to be performed twice: one prior to medication intake and again postintake to measure ON and OFF states. While there is no designated time of day for these tests, participants are instructed to consistently follow this premedication and postmedication schedule.

The tasks included in the Mobistudy app for this study are:

- ▶ Questionnaires:
  - Beck's depression inventory-II (BDI-II).[22] This is used to quantify depression.
  - Parkinson's disease sleep scale (PDSS-2).[23] Used to quantify sleep disorder.
  - Non-Motor Symptoms Questionnaire (NMSQuest).[24] Measures non-motor symptoms.
  - Parkinson's Disease Questionnaire (PDQ-8).[25] Short scale to assess motor and non-motor symptoms in daily life.
  - Sleep quality questionnaire. Based on,[26] is a custom daily short questionnaire to assess sleep quality (full questions list in online supplemental appendix 1).
  - User Version of the Mobile Application Rating Scale (uMARS).[27] Measures usability and the quality of the app.
  - Technology Acceptance Model (TAM).[28] Estimates long-term usage of the app.
- ▶ Timed up and go test (TUG test): Participants are instructed to stand up from a chair, walk 3 meters, turn around, return to the chair, and sit down. The mobile phone, secured in a waist belt around the hip, records data from inertial sensors (accelerometer and gyroscope) and estimates the total time required to complete the task.
- ▶ Vocalization test: Participants record their voice during a vocalization exercise guided by the app. The exercise consists of pronouncing the sustained vowels "a", "i" and "u" for as long as possible.
- ▶ Hold the phone test (HTP test): Participants are instructed to hold the phone for 60 seconds while the phone records inertial sensors data (acceleration and orientation). The task is repeated for each hand in

three positions: resting the hand holding the phone in the lap while sitting, extending the arm at shoulder's height, and moving the arm from outstretched to touching the nose repeatedly.

► Finger tapping test (FT test): Participants are instructed to rapidly tap two buttons on the screen using the index and middle fingers.

► Drawing test: Participants are instructed to draw a square and a squared spiral on the screen by tracing a marked line with their index finger.

In addition to the app, a wrist-worn actigraphy device (GENEActiv, Activinsights, UK) is used to collect continuous three-dimensional acceleration data in free-living conditions. A gait analysis device (GWALK, BTS Bioengineering, Italy) is used to gather reference gait analysis data during the TUG test in clinic, at enrolment and conclusion of the study. The GENEActiv device is set to capture data at a sample rate of 25 Hz. While 10 Hz is considered sufficient for wrist-based acceleration in day-to-day activities,[29] this study adheres to the Nyquist theorem by using at least twice the maximum frequency of tremor.[30]

While aiming at including a wide range of symptomatology, in this first feasibility study, we decided to focus on motor symptoms, also given the nature of the sensors employed. We decided to exclude cognitive function, as this would have required the implementation of additional interactive features on the Mobistudy app that were not justified by the scope of the research.

Participants are approached during their regular visits to the ASCT clinic. If they give informed consent to participate, they receive an introduction to the study, instructions, a waist belt and an actigraphy device. The Mobistudy app is downloaded and installed on their smartphone, and participants create an account. A series of demo tests using Mobistudy is available for participants to practice and ensure understanding before the study starts. Each test includes in-app instructions. A TUG test, performed with the GWALK gait analysis reference device, and a full MDS-UPDRS assessment by a neurologist are conducted during enrollment.

During the 8-week at-home period, participants complete tests and answer questionnaires using their phones and the Mobistudy app (illustrated in figure 1). Participants are contacted on day 10 and on day 30 either via email or phone call to check compliance and the presence of any technical or practical issues.

On completing of the study, participants attend a second clinical visit, during which they perform a TUG test with GWALK, undergo a full MDS-UPDRS assessment and complete an interview. The semistructured interview is designed to complement questions asked in the usability questionnaires and covers motivations for participation, usability, user-friendliness, acceptance, perceptions of trust and suggestions for improvements (full schedule in online supplemental appendix 2). A researcher conducts the interview with the participant either in the clinic or over the phone

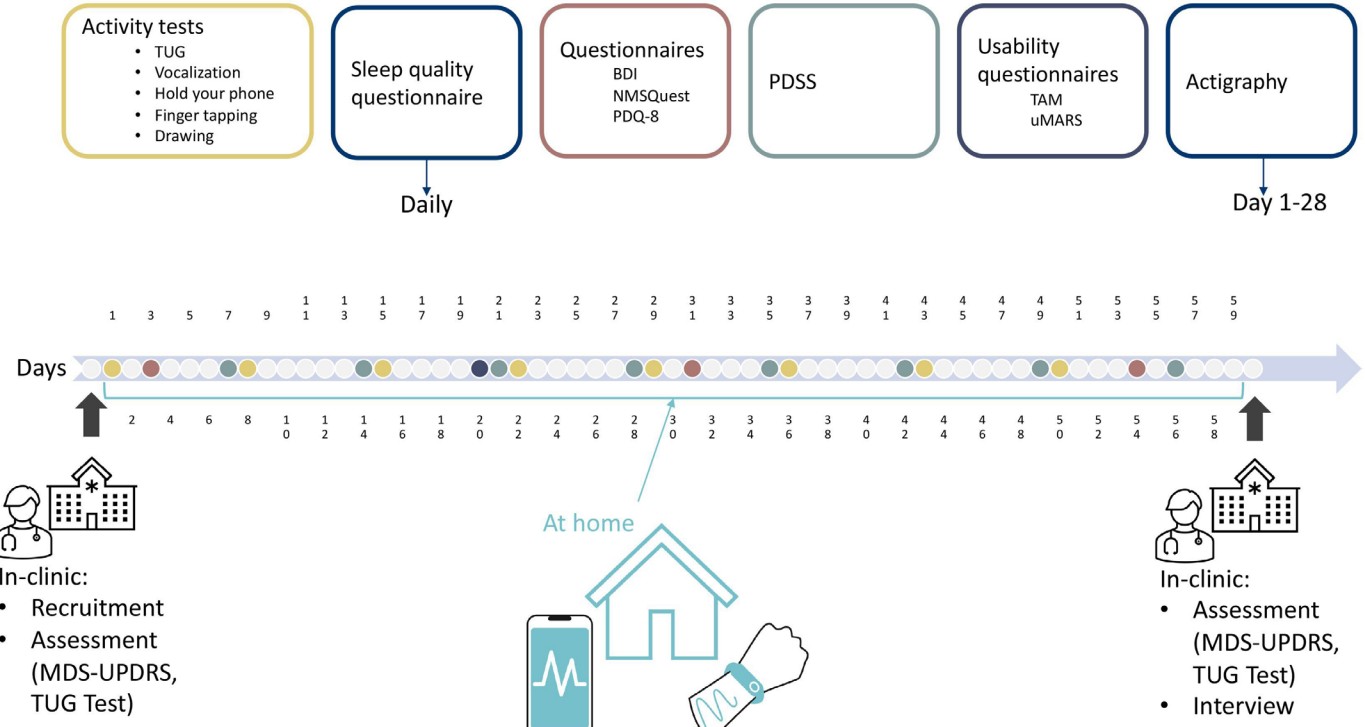

**Figure 1** Timeline of the study depicting assessments, tests and questionnaires, when and where they take place. BDI, Beck's Depression Inventory; NMSQuest, Non-Motor Symptoms Questionnaire; PDQ-8, Parkinson's Disease Questionnaire; PDSS, Parkinson's Disease Sleep Scale; TAM, Technology Acceptance Model; TUG, Timed Up and Go; uMARS, User Version of the Mobile Application Rating Scale; MDS-UPDRS, Movement Disorder Society-Unified Parkinson Disease Rating Scale.

if consent is granted. Interviews are expected to last no more than 30 min and are audio recorded.

## Data management

Clinical data, including patients' demographics (age, sex, year of diagnosis of PD, year of onset of first symptoms, deep brain stimulation device status, any PD-related surgery, medication regime and comorbidities) as well as in-clinic results (evaluated MDS-UPDRS and TUG test), are collected by the ASCT. These data are stored on an encrypted disk on a computer managed by the research team.

App-based data is collected using the Mobistudy platform.[21] This includes participants' profiles—name, surname, date of birth, sex, height, weight, long-term diseases, long-term medications—questionnaire responses and sensors data. Sensors data includes acceleration and orientation during the TUG test and HTP test, voice recordings collected during the vocalisation test, timestamps of taps during the FT test and finger coordinates during the drawing test. Metadata including time and date of when a data collection task was executed, and phone model and make, are also recorded. These data are stored on servers owned and managed by Malmö University in Sweden, they are encrypted during the transmission and at rest and are accessed only by researchers under strict rules.

The GENEActiv actigraphy device stores acceleration data internally. On the study's completion, participants return the device, the researcher charges it, extracts and saves the data on the study computer marked with the patient identifier.

Interviews of participants are conducted by researchers in person (when possible) or over the phone and recorded for later analysis with patient's consent.

Malmö University acts as central repository for data, including data collected via Mobistudy and by ASCT. Data is shared among researchers using the Box cloud storage service managed by the University and employing the Cryptomator (https://cryptomator.org/) encryption software.

Collaborators from the University of Edinburgh receive access to pseudonymised data such as general demographic information, MDS-UPDRS assessments, audio data from the vocalisation test and the data from the GENEActiv wrist-wearable device. These data are shared through a separate shared folder, containing no personal information.

Data will be removed from the Mobistudy servers on completion of data collection but retained on researchers' computers during analysis. An archive will be kept for 10 years in accordance with current regulations.

## Quality assurance

Data are reviewed monthly to ensure comprehensive and accurate recording at the clinic and through the app. Every 2 months, interim data analyses are also put in place to allow in-depth quality checks and possible bugs in the

software. Non-compliant participants are contacted by the clinical team to explore reasons and propose solutions.

## Data analysis plan

Quantitative data will be analysed using statistical, signal processing and machine learning methods, while qualitative data will be evaluated based on the theoretical foundations underlying the questionnaires and interview questions.

### Validity of measurements collected by mobile phones and wearables

From the data collected from activity tests and the wrist-wearable device, we will extract features and train machine learning models to infer the severity of the disease using the MDS-UPDRS score as a reference. This includes measuring time needed to complete the TUG test, finger dexterity in the FT test, tremor in the HTP and the drawing tests, and assessing voice distortions in the vocalisation test.[31] The main metric for assessing validity will be correlation between features and/or inferred scores and actual scores. We will associate the measurements to the MDS-UPDRS closest in time. The variability of fluctuating symptoms will be addressed by exploring how averaging features, including before and after medication and over the weeks, affects the correlation. The data collected by the wearable will also offer a reference to understand if and in what way symptoms have evolved over time.

As targets, we will employ both summative scores, for example, for part 3 (motor symptoms), which will be compared with aggregated results from all the tests, and specific items, when those can be mapped to the result of a specific test, for example, finger tapping with item 3.4.

Data acquired by the GWALK gait analysis device during the TUG tests in clinic will be employed as a reference to validate the algorithms used to analyse the data collected by the app in the same test.

Algorithms will be implemented to assess tremor, bradykinesia and sleep quality from the actigraphy data obtained from the wearable device. Sleep-related questionnaires will serve as a reference for sleep quality assessment.

Various machine learning classifiers such as AdaBoost, random forest, multilayer perceptron, support vector machine, XGBoost and similar algorithm for regression analysis will be trained to predict symptoms scores. Regression models will be used to predict the scale of the MDS-UPDRS whereas classifiers will be used to predict scores binned into boarder categories (eg, no symptoms, low symptoms, high symptoms). A preliminary proof of feasibility has been performed through a regression analysis to infer disease severity[32] based on a dataset with self-reported part of MDS-UPDRS components collected in the mPower study.[33] This offered insights into the anticipated quality from data collected in free-living conditions. Importantly, a correlation between data collected from smartphones and symptoms' severity was confirmed but only after removing a substantial number of participants

who were considered unreliable. This study aims to address those deficiencies by selectively recruiting participants with clinically diagnosed PD.

Algorithms for classification and regression analysis will be compared on different evaluation metrics including common performance metrics such as accuracy, sensitivity, specificity and area under the curve-receiver operating characteristic (AUC-ROC) for classifiers. For regression models, metrics like mean squared error, mean absolute error and R square measures will be used.

Feature selection (FS) will be performed, and the best predictive features will be identified. Methods mentioned in,[34] Relevance, Redundancy, and Complementarity Trade-off, minimal-redundancy-maximal-relevance (mRMR Peng), correlation-based FS and recursive feature elimination will be explored, as well as dimensionality reduction techniques such as principal component analysis.[35] To assess the performance of different machine learning algorithms, k-fold cross-validation and leave-one-participant-out cross-validation method will be used, in order to account for participant identity confounding.

### Compliance, usability and acceptance

Participants' compliance is measured using activity logs from which we can derive the number of tests performed and questionnaires answered.

Usability and long-term acceptance of the app will be assessed by computing descriptive statistics about the quantitative answers given to the questionnaires and performing a thematic analysis of the interviews based on the theoretical foundations that defined the questions.

### Strengths and limitations

Most existing digital systems for PD assessment focus on one symptom, for example, tremor, or cognitive decline. Our proposed approach is multidimensional, aiming at a more holistic view of the participants' status. Another important strength of this study is that sensors are used to measure symptoms objectively and reliably while performing motor tasks and within a daily life environment, thus allowing frequent measurements.

Limitations of the study include the sample size and sufficiently representing all disease severity levels. Patients with severe PD are excluded from the study, which could impact the generalisability of the results. Furthermore, given the home-based setting of this study, the quality of the data may be affected by the participants' degree of participation and compliance to the study protocol.

**Acknowledgements** We would like to thank our collaborators at Region Stockholm, Karolinska Institute and University of Edinburgh. We would like to thank and acknowledge the financial support provided by Mats Paulsson Foundation and Malmö University. Most importantly, we would like to thank our participants, who have given their time and shared their valuable experiences in being part of this study. Finally, we would like to thank the Internet of Things and People Research Center for their technical and financial support.

**Contributors** GY: methodology, software, data curation, formal analysis, investigation, writing—original draft, visualisation. DS: conceptualisation, methodology, software, writing—review and editing, supervision, project administration, funding acquisition. CMO: conceptualisation, methodology, writing—

review and editing, supervision, funding acquisition. MVW: investigation, resources, data curation, writing—review and editing. AT: conceptualisation, methodology, writing—review and editing, supervision, funding acquisition. PS: conceptualisation, methodology, resources, writing—review and editing, supervision, funding acquisition.

**Funding** Mats Paulsson foundation, grant number: NA. Internet of Things and People (IOTAP) research center, grant number: NA. Malmö University, grant number: NA.

**Competing interests** None declared.

**Patient and public involvement** Patients and/or the public were not involved in the design, or conduct, or reporting, or dissemination plans of this research.

**Patient consent for publication** Not applicable.

**Provenance and peer review** Not commissioned; externally peer reviewed.

**ORCID iD**
Gent Ymeri http://orcid.org/0000-0002-7102-083X

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
