## [Reviewer comments · BMJ Open]

ARTICLE DETAILS

TITLE (PROVISIONAL)	Quantifying Parkinson's disease severity using mobile wearable devices and machine learning; The ParkApp pilot study protocol
AUTHORS	Ymeri, Gent; Salvi, Dario; Olsson, Carl Magnus; Wassenburg, Myrthe; Tsanas, Athanasios; Svenningsson, Per

VERSION 1 – REVIEW

REVIEWER	Prell, Tino Halle University Hospital, Department of Geriatrics
REVIEW RETURNED	21-Aug-2023

GENERAL COMMENTS	I only have a few minor comments: I miss a statement and assessment of cognitive function, because cognitive problems are quite common in PD. Inclusion/exclusion: - Consider to define a threshold for cognitive function. Define age range for inclusion. - "Must have mild or moderate symptom" What do you mean? Motor or NMS or both? Please detail. Outcome: "to provide a more holistic understanding of Parkinson's symptoms?" This is a bit unspecific to use it as a outcome measure. "check compliance, the presence of any technical issues and asked for feedback on user-friendliness of the app." How exactly do you measure adherence and user-friendliness (provide the planned questions/interviews). Analyses: Do you use the MDS-UPDRS sum score or only subscores as reference? Explain the rationale to use sum score or subscores, as the used independent variables cover a wide range of motor symptoms and NMS. Why do you use a plethora of machine learning methods? How do you want to compare them and are you going to publish findings from each method? Are they all suitable for N = 40? How do you handle motor fluctuations as these might influence the obtained measures? Consider to include other measures (e.g. loneliness, social exclusion) as these are common and can have large impact on QoL in PD.
--

VERSION 1 – AUTHOR RESPONSE

Reviewer: 1

Comments to the Author:

I only have a few minor comments:

I miss a statement and assessment of cognitive function, because cognitive problems are quite common in PD.

- We thank the reviewer for this comment. We recognize that cognitive function is an interesting subject and that mobile technologies have already been tested both for measurement and treatment in that regard. However, at the time of designing the study, we decided to not include cognitive function in order to reduce the complexity. We have added a sentence on page 6 regarding this point:
- "While aiming at including a wide range of symptomatology, in this first feasibility study, we decided to focus on motor symptoms, also given the nature of the sensors employed. We decided to exclude cognitive function, as this would have required the implementation of additional interactive features on the Mobistudy app that were not justified by the scope of the research."

Inclusion/exclusion:

- Consider to define a threshold for cognitive function. Define age range for inclusion.
- Neither for age nor cognitive function, we have been officially including a range in the criteria, however, as stated on page 4 "an ad-hoc approach is adopted depending on the circumstances". This includes estimating if the patient is able to use the phone and perform the tests independently. Among exclusion criteria, we have listed "Not able to use a smartphone", which would exclude cases of more severe cognitive decline that would affect how data is collected. We have added the word "independently" to clarify that help from a carer would not suffice.
- "Must have mild or moderate symptom" What do you mean? Motor or NMS or both? Please detail.
- We are considering both motor and non-motor symptoms when recruiting, so we have rephrased the criterium to: 'Must have mild or moderate motor and non-motor symptoms'

Outcome: "to provide a more holistic understanding of Parkinson's symptoms?" This is a bit unspecific to use it as a outcome measure.

- We recognize that the way the research question was formulated was a bit vague. We have decided to highlight the focus on validity and thus rephrased the research question to "Are the multimodal measurements collected by mobile phones and wearable devices valid for estimating Parkinson's disease symptoms?"

"check compliance, the presence of any technical issues and asked for feedback on user-friendliness of the app." How exactly do you measure adherence and user-friendliness (provide the planned questions/interviews).

- The answer to this question is available on page 9. Adherence is measured at the end of the study by evaluating the rate of completion of the assigned tasks. User aspects, including usability, are assessed through questionnaires (see page 5, reference to uMARS, TAM) and an interview (page 6). We have added the interview schedule in Appendix 2, as for standard questionnaires, references should suffice.

The quoted paragraph in the question refers to the fact that participants are contacted between clinical visits to ensure that they have no major problems in conducting the study, including technical issues, use of the app and wearables, availability, and other practicalities. We have rephrased this sentence in the attempt to improve clarity: "Participants are contacted at day ten and at day 30 either via email or phone call to check compliance and the presence of any technical or practical issues".

Analyses:

Do you use the MDS-UPDRS sum score or only subscores as reference? Explain the rationale to use sum score or subscores, as the used independent variables cover a wide range of motor symptoms and NMS.

- We use subscores and cumulative scores for various components of the MDS-UPDRS. This approach is driven by our objective to map activity tests and questionnaires with specific items in the MDS-UPDRS. For example, the use of the sum score, in MDS-UPDRS Part 3, is primarily intended to correlate all activity tests with Part 3, emphasizing motor symptoms. At the same time, we aim to map the subscore of the finger-tapping item in MDS-UPDRS to the finger-tapping task. We strive to optimize the mapping of our questionnaire questions and activity tests with the respective subscores wherever feasible. We have added the following to page 8: "As targets, we will employ both summative scores, for example for Part 3 (motor symptoms), which will be compared with aggregated results from all the tests, and specific items, when those can be mapped to the result of a specific test, for example: finger tapping with item 3.4."

Why do you use a plethora of machine learning methods? How do you want to compare them and are you going to publish findings from each method? Are they all suitable for N = 40?

- As customary in computer science, different ML algorithms will be tested to determine which performs best according to appropriate evaluation metrics depending on whether the problem is treated as a classification or regression (accuracy, sensitivity, specificity, AUC, etc.). Evaluation metrics are mentioned in the subsection, now rephrased as "Validity of measurements collected by mobile phones and wearables", on page 8. The methods were chosen because suitable for this number of patients, as also shown in the research we mention in reference 32.

How do you handle motor fluctuations as these might influence the obtained measures?

- We recognise that fluctuation poses a challenge for the analysis. For that, we are planning to relate measurements to the closest MDS-UPDRS score and average features over time. We have added the following on page 8: "We will associate the measurements to the MDS-UPDRS closest in time. The variability of fluctuating symptoms will be addressed by exploring how averaging features, including before and after medication and over the weeks, affect the correlation. The data collected by the wearable will also offer a reference to understand if and in what way symptoms have evolved over time."

Consider to include other measures (e.g. loneliness, social exclusion) as these are common and can have large impact on QoL in PD.

- This is a relevant suggestion and very much appreciated, however, as the study has already commenced, it would be hard to include additional measures at this stage. We hope that we will be able to partially capture these aspects, or at least correlate, through the existing questionnaires (BDI-2, NMSQuest, PDQ-8). Future work can be directed at extending the measurement of non-motor symptoms in PD.

Additional change: While originally, we planned for 40 participants, as the study went on, we took the decision that 30 would be enough, so we are correcting this to avoid confusion in future publications.

VERSION 2 – REVIEW

REVIEWER	Prell, Tino Halle University Hospital, Department of Geriatrics
REVIEW RETURNED	27-Nov-2023
GENERAL COMMENTS	Apparently the study is already running and therefore nothing more can be done about the design or the assessments (which in turn somewhat limits the usefulness of my review). Therefore, I agree with the changes made. Ultimately, it is the authors' problem to get the data published if there are methodological weaknesses (number of cases now arbitrarily

	reduced to 30; no assessment of cognition, although many patients have dementia, etc.). So from my side, I have no further requests for changes.
--	--